# LEARNING LAGRANGIAN FLUID DYNAMICS WITH GRAPH NEURAL NETWORKS

## ABSTRACT

We present a data-driven model for fluid simulation under Lagrangian representation. Our model uses graphs to describe the fluid field, where physical quantities are encoded as node and edge features. Instead of directly predicting the acceleration or position correction given the current state, we decompose the simulation scheme into separate parts - advection, collision, and pressure projection. For these different reasoning tasks, we propose two kinds of graph neural network structures, node-focused networks, and edge-focused networks. By introducing physics prior knowledge, our model can be efficient in terms of training and inference. Our tests show that the learned model can produce accurate results and remain stable in scenarios with a large number of particles and different geometries. Unlike many previous works, further tests demonstrate that our model is able to retain many important physical properties of incompressible fluids, such as minor divergence and reasonable pressure distribution. Additionally, our model can adopt a range of time step sizes different from ones using in the training set, which indicates its robust generalization capability.

## 1 INTRODUCTION

For many science and engineering problems, fluids are an essential integral part. How to simulate fluid dynamics accurately has long been studied by researchers and a large class of numerical models have been developed. However, computing high-quality fluid simulation is still computationally expensive despite the advances in computing power. Also, the time of calculation usually increases drastically when the resolution of the simulating scene scales up. A common way to alleviate computing costs is using a data-driven model. Recent progress in the machine learning domain opens up the possibility of employing learning algorithms to learn and model fluid dynamics.

In this paper, we propose a graph-based data-driven fluid dynamics model (Fluid Graph Networks, FGN), which consists of simple multi-layer perceptron and graph inductive architectures. Our model predicts and integrates forward the movement of incompressible fluids based on observations. Compared to previous works in this domain (Ummenhofer et al., 2020; Sanchez-Gonzalez et al., 2020), our model enjoys traceability of physical properties of the system, like low velocity-divergence and constant particle density, and it can predict reasonable pressure distribution. Experiments demonstrate that our model can remain stable and accurate in long-term simulation. Although our model is entailed and customized for fluid simulation, it can be extended to simulation of other dynamics under the Lagrangian framework, as it takes universal features (positions, velocities, particle density) under the Lagrangian framework as input.

## 2 RELATED WORKS

Our model is built upon the Lagrangian representation of fluid, where continuous fluids are discretized and approximated by a set of particles. The most prominent advantage of the Lagrangian method is that the particle boundary is the material interface, which makes boundary conditions easy to impose, especially when the material interface is large and changing violently. A well-known Lagrangian method is Smooth Particle Hydrodynamics (SPH)(Monaghan, 1988). SPH and its variants are widely used in the numerical physic simulation, especially fluid dynamics under various environments. Particle-based fluid simulation (Müller et al., 2003) introduces SPH model to simulate fluids and

generate realistic visual effects. Moving particle semi-implicit method (MPS) (Koshizuka and Oka, 1996) markedly improves the accuracy and stability of incompressible fluid simulation by introducing a pressure projection procedure that emulates Eulerian grid-based methods. Weakly compressible SPH (WSPH)(Becker and Teschner, 2007) introduces equation of state to model the pressure during the simulation. Predictive-corrective incompressible SPH (Solenthaler and Pajarola, 2009) and divergence-free SPH (Bender and Koschier, 2015) use iterative method to improve the accuracy of pressure calculation in incompressible flow simulation.

Modeling fluid dynamics in a data-driven way has been explored and studied by many researchers. With advances in machine learning algorithms, many data-driven models employing machine learning algorithms have been built. Ladický et al. (2015) reformulate the Navier-Stokes equation as a regression problem and build a regressor using random forest, which significantly improves the calculation efficiency. Tompson et al. (2016), Xiao et al. (2020) learn the pressure projection under the Eulerian framework with a convolutional neural network, which accelerates the fluid simulation. Wiewel et al. (2018) bring significant speed-up by learning a reduced-order representation and predicting the pressure field with an LSTM-based model. Morton et al. (2018) learn the dynamics of airflow around a cylinder based on Koopman theory. de Avila Belbute-Peres et al. (2020) predict fluid flow by combining grid-based method with graph convolutional neural networks.

Learning and reasoning particle dynamics under graph representation has the following benefits and conveniences. First, particle-based methods model physics phenomena as interactions between particles within a local area. This imposes an inductive bias for learning under the Lagrangian framework: dynamics have a strong locality. The locality of unstructured data under Lagrangian representation can be captured by aggregation operation on graphs, such as GCN and other variants (Kipf and Welling, 2016; Hamilton et al., 2017). Second, unlike Eulerian grid-based methods, Lagrangian particle-based methods do not have explicit and structured grid, which makes standard Convolutional Neural Network (CNN) cannot be directly applied to particles without feature processing (Wang et al., 2018; Ummenhofer et al., 2020). Third, many dynamics are based on pairwise relation between particles, like collision, which can be easily interpreted as edge attributes of a graph. Given these factors, recently there have been a rich class of works that use graph neural networks (Scarselli et al., 2009) to learn and reason about underlying physics of interacting objects and particles. (Battaglia et al., 2016; Chang et al., 2016; Sanchez-Gonzalez et al., 2018; Li et al., 2018; Mrowca et al., 2018)

## 3 MODEL

### 3.1 FLUID DYNAMICS

The governing equation for incompressible fluids is the Navier-Stokes equation and the continuity equation as follows (Batchelor, 2000):

$$\frac{D\mathbf{u}}{Dt} = -\frac{\nabla p}{\rho} + \nu\nabla^2\mathbf{u} + \mathbf{g}, \tag{1}$$

$$\nabla \cdot \mathbf{u} = 0. \tag{2}$$

To describe the fluid field, there are two kinds of systems, Eulerian and Lagrangian ones. In this work, we adopt a Lagrangian system. A common method to solve the Navier-Stokes equation and discretize fluids under the Lagrangian framework is Smooth Particle Hydrodynamics (SPH) method (Monaghan, 1988), where physical quantities at an arbitrary point in the space are approximated by the states of nearby particles.

In SPH, an arbitrary scalar (or vector) field $A(\mathbf{r})$ at location $\mathbf{r}$ can be represented by a convolution:

$$A(\mathbf{r}) = \int A(\mathbf{r}') W(|\mathbf{r} - \mathbf{r}'|, h) \, dV(\mathbf{r}'), \tag{3}$$

where $W$ is weighting function or smooth kernel as defined in SPH, $h$ is the smoothing length, which defines the range of particles to be considered and $V(\mathbf{r}')$ is the volume at $\mathbf{r}$. Numerically, the interpolation can be approximated by replacing the integration with a summation.

Based on this model, equation equation 1 and equation 2 can be discretized. The discrete equation system is usually solved under a predictor-corrector scheme, prediction based on advection and correction based on physical properties (such as divergence-free constraint).

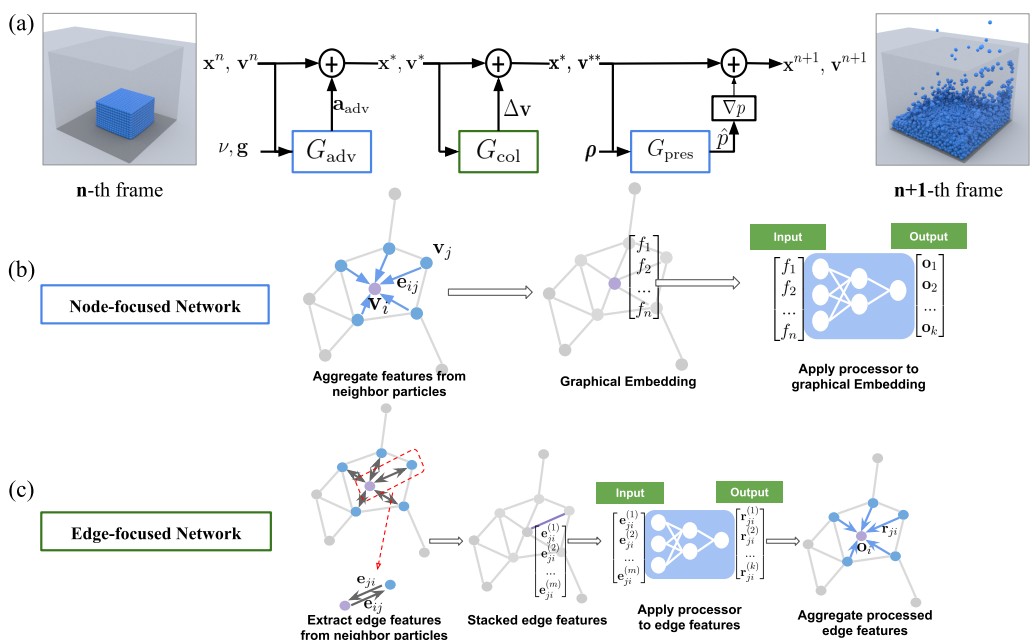

Figure 1: **(a)** Schematic of our FGN model. $G_{\text{adv}}$ applies the effect of body force and viscosity to the fluids. $G_{\text{pres}}$ predicts the pressure. $G_{\text{col}}$ handles collision between particles. **(b)** In a node-focused network, input are represented as node features and then passed to a shared processor. Advection and pressure impose influence on each particle separately, thus they can be interpreted as messages lying on each node and predicted via node-focused networks. **(c)** In an edge-focused network, input are represented as edge features of a directed graph, the edge features are then passed to a shared processor. Collision is a pairwise effect, which can be easily represented as edge attributes between each pair of particles. Hence we use the edge-focused network to predict collision.

## 3.2 MODEL

Fluids are time-dependent dynamical systems, where location of particles, $\mathbf{r}$, is described by equation of form: $d\mathbf{r}/dt = f(\mathbf{r})$. When building a data-driven model to learn and solve this system, we assume the system is Markovian, that is, the state of the system at time step $n + 1$ depends only on the state of the previous time step $n$. The update mechanism in our model can be represented as:

$$\{\mathbf{x}^{n+1}, \mathbf{v}^{n+1}\} = G_\theta\left(\{\mathbf{x}^n, \mathbf{v}^n\}\right). \tag{4}$$

Here $\{\mathbf{x}^n, \mathbf{v}^n\}$ denotes the positional information and velocity of fluid field at time step $n$. Data-driven model $G_\theta$, parameterized by $\theta$, maps the state of time step $n$ to time step $n + 1$.

In order to build a robust and accurate data-driven model, the structure of our model is physic-informed, which enables the model to give interpretable output without losing many physical properties of the system. In general, our model mimics the predictor-corrector scheme and includes three parts, advection net, collision net, and pressure net. They can be divided into two types of graph networks (GN) according to the network structure (Battaglia et al., 2018). Specifically, advection net and pressure net are node-focused graph networks, while collision net is edge-focused networks. As each of these networks has a specific task and different output, they are trained on different data separately.

**Node-focused Graph Network** Advection net is responsible for the prediction of advection effect and pressure net is responsible for pressure projection. Considering a particle $i$, the node-focused graph network first aggregates node features from neighbor particles $\{v_j | \forall j \in \mathcal{N}(i)\}$ and output node embedding $\mathbf{f}_i$. The embedding $\mathbf{f}_i$ will then be passed to a processor $g_R$. $g_R$ will predict the desirable physical quantities $o_i$ (i.e. acceleration $\mathbf{a}$ in advection net and pressure $p$ in pressure net).

The whole message passing procedure can be defined as:

$$o_i = g_R \left( g_A \left( V_i \right) \right), V_i = \{v_j\}_{\forall j \in \mathcal{N}(i)}. \tag{5}$$

**Edge-focused Graph Network**   To prevent particle penetration and increase model stability, we propose a graph network model that is responsible for predicting the effect of collision. As the relative position and relative velocity will have different signs with different observation perspective (i.e. relative velocity $v_{ij} = -v_{ji}$), thus the graph in collision net is directed. In collision net, relative features, $e_{ij}$ (relative positions, relative velocities between particle $i$ and $j$), are passed to processor $f_R$ as edge features. The processor will output the edge embedding $r_{ij}$ between each pair of nodes. Lastly, edge embedding $r_{ij}$ is aggregated via aggregator $f_A$, to gather the influence from all nearby particles and predict an overall effect $o_i$ on the center particle $i$. The whole process is defined as:

$$o_i = f_A \left( f_R \left( E_i \right) \right), E_i = \{e_{ij}\}_{\forall j \in \mathcal{N}(i)}. \tag{6}$$

The advantage of using relative position and velocity instead of global ones as input features is that this explicitly imposes a spatial invariance to the network, given that collision between two particles is invariant to the global positions they are at.

## 4   IMPLEMENTATION

We adopt the numerical model in SPH to evaluate the physical quantities like particle density and differential operators like gradient. To construct graph representation for particles, we establish edges between particles within the control radius.

### 4.1   UPDATE SCHEME

In general, given the state (position $\mathbf{x}^n$ and velocity $\mathbf{v}^n$) of the current time step $n$, we derive the state of next time step $n+1$ by passing the state information through advection net, collision net, and pressure net sequentially. The input features to advection net are positions and velocities of particles, $[\mathbf{x}^n, \mathbf{v}^n]$, along with $\mathbf{g}$, which indicates the external body force per mass of fluid, and viscosity parameter $\nu$, which denotes the magnitude of fluid viscosity. The advection net predicts acceleration of particles:

$$\mathbf{a}^{\mathrm{adv}} = G_{\mathrm{adv}} \left( \mathbf{x}^n, \mathbf{v}^n, \mathbf{g}, \nu \right), \tag{7}$$

and updates the state of fluid particles to an intermediate state $[\mathbf{x}^*, \mathbf{v}^*]$.

$$\mathbf{v}^* = \mathbf{v}^n + \mathbf{a}^{\mathrm{adv}} \Delta t, \tag{8}$$
$$\mathbf{x}^* = \mathbf{x}^n + \mathbf{v}^* \Delta t. \tag{9}$$

Where $\mathbf{a}^{\mathrm{adv}} = [\mathbf{a}_1^{\mathrm{adv}}, ..., \mathbf{a}_N^{\mathrm{adv}}]$, $\mathbf{x}^n = [\mathbf{x}_1^n, ..., \mathbf{v}_N^n]$, $\mathbf{v}^n = [\mathbf{v}_1^n, ..., \mathbf{v}_N^n]$ for particles $i \in \{1, .., N\}$. We will use the same notation throughout illustration.

The collision net takes relative positions and velocities between particles, $[\mathbf{x}_i^* - \mathbf{x}_j^*, \mathbf{v}_i^* - \mathbf{v}_j^*]$ as input, and predicts correction to the velocity,

$$\Delta \mathbf{v} = G_{\mathrm{col}}(\mathbf{x}_r^*, \mathbf{v}_r^*), \tag{10}$$

where $[\mathbf{x}_r^*, \mathbf{v}_r^*]$ denotes the relative position and velocity in intermediate state. The velocity is then updated with predicted correction:

$$\mathbf{v}^{**} = \mathbf{v}^* + \Delta \mathbf{v}. \tag{11}$$

The updated intermediate position and velocity are taken as input by the pressure net, along with particle number density $\rho$.

$$\hat{\mathbf{p}} = G_{\mathrm{pres}}(\mathbf{x}^*, \mathbf{v}^{**}, \rho). \tag{12}$$

The state of fluid field is then updated to next time step $n+1$,

$$\mathbf{v}^{n+1} = \mathbf{v}^{**} - \frac{\nabla \hat{\mathbf{p}}}{\rho_c} \Delta t, \tag{13}$$

$$\mathbf{x}^{n+1} = \mathbf{x}^* + \mathbf{v}^{n+1} \Delta t. \tag{14}$$

Where $\hat{\mathbf{p}} = [\hat{p}_1, ..., \hat{p}_N]$, $\boldsymbol{\rho} = [\rho_1, ..., \rho_N]$, $\rho_c$ is the density parameter of fluid. Predicting pressure of fluid field using particle density and velocity is based on the observation that advection will incur a temporary compression on fluid body, which means fluid density has changed. Therefore the goal of pressure net is to impose a pressure projection to mitigate these deviations.

During the above calculation, the global positional information is only used to construct graph on fluid particles and will not be passed into aggregator and processor as features. The relative position and particle density are normalized before input.

## 4.2 NETWORK ARCHITECTURES

For node-focused graph network, to derive a smooth response of the field with respect to spatial location, the aggregation from layer $l - 1$ to $l$ is defined as:

$$\mathbf{a}_i^{(l)} = \frac{\sum \mathbf{f}_j^{(l-1)} W\left(|\mathbf{r}_i - \mathbf{r}_j|, h\right)}{\sum W\left(|\mathbf{r}_i - \mathbf{r}_j|, h\right)} + \mathbf{f}_i^{(l-1)}, \forall j \in \mathcal{N}(i), \tag{15}$$

$$\mathbf{f}_i^{(l)} = \sigma\left(\mathbf{W} \cdot \mathbf{a}_i^{(l)}\right), \tag{16}$$

where the aggregator sums up the features $\mathbf{f}_j^{(l-1)}$ from neighbor vertices $\{v_j | j \in \mathcal{N}(i)\}$ using smooth kernel as weight function, and here self-connection is added to every vertex. Linear transformation $\mathbf{W}$ and non-linear transformation $\sigma$ are then applied to the aggregated features. In practice we found that two layer of aggregations is enough for the model to produce reasonably accurate output (Adding more aggregation layers does not bring in significant improvements).

As for the edge-focused network, the aggregation is simply defined as:

$$\mathbf{a}_i = \sum_{j \in \mathcal{N}(i)} \mathbf{W} \cdot \mathbf{r}_{ji}, \tag{17}$$

where $\mathbf{r}_{ji}$ is edge feature processed by processor, $\mathbf{W}$ is a linear transformation matrix. The aggregation in the edge-focused network is at the last layer, so no non-linearity is included here.

The processors in both networks are implemented as shared MLP, where they are shared among nodes or edges depending on network types (e.g. in the node-focused network, the processor is an MLP shared among each node). In a node-focused network, the processor has three hidden layers, with the input node embedding $\mathbf{f}$ of size 128. In an edge-focused network, the processor has four hidden layers, with input edge attributes $[\mathbf{x}_r, \mathbf{v}_r]$ of size 6.

## 5 EXPERIMENTS

### 5.1 TRAINING

**Dataset** We use the Moving Particle Semi-implicit method (MPS) (Koshizuka and Oka, 1996) with an improved pressure solver (Lee et al., 2011) to generate high-fidelity simulation data of incompressible flow. MPS is a numerical method based on SPH which prioritizes accuracy over calculation speed. It enforces the incompressibility of the fluid field by solving the pressure Poisson equation. We created 20 scenes by randomly placing fluid blocks, solid obstacles, initializing fluid particles with random velocity (See A.2 for full detail of training dataset settings and training strategy).

**Loss Function and Optimization** We train three networks, advection net, collision net, and pressure net separately. Each network is trained in a supervised way by optimizing the mean squared error between prediction $\hat{y}$ and ground truth $y$.

$$L = \frac{1}{N} \sum_{i=1}^{N} ||y_i - \hat{y}_i||_2^2 \tag{18}$$

We normalize particle density before inputting into the pressure net, which accelerates and stabilizes training. In the processor, we add LayerNorm (Ba et al., 2016) after activation to each layer (except

| Case | Model | Density error | | Velocity divergence | | Average Chamfer distance (mm) | |
|------|-------|------|------|------|------|------|------|
| | | Mean | Max | Mean | Max | Mean | Max |
| Dam Collapse | FGN (this work) | **0.0461** | **0.0900** | **0.0195** | **0.0280** | **24.2** | **30.1** |
| | GNS | 0.0898 | 0.3387 | 0.0210 | 0.0311 | 26.1 | 33.1 |
| | CConv | 0.1811 | 0.2947 | 0.0338 | 0.1700 | 31.4 | 44.0 |
| | Ground Truth | 0.0380 | 0.0710 | 0.0190 | 0.0268 | - | - |
| Water Fall | FGN (this work) | **0.0541** | **0.1350** | **0.0207** | 0.0431 | **24.8** | **29.6** |
| | GNS | 0.1035 | 0.3512 | 0.0223 | **0.0399** | 25.6 | 31.8 |
| | CConv | 0.2783 | 0.4121 | 0.0668 | 0.2882 | 40.4 | 61.5 |
| | Ground Truth | 0.0429 | 0.0966 | 0.0196 | 0.0398 | - | - |

Table 1: Quantitative accuracy analysis. For resolution, the dam collapse case contains about 10k particles and water fall case contains about 80k particles. We report the mean and maximum value of density error and velocity divergence over the whole simulation sequence. We also report the average Chamfer distance between the results of each model and ground truth data.

for the output layer). The parameters of the model are optimized with Adam (Kingma and Ba, 2014) optimizer. We implement the model in PyTorch. All the training and experiments are mainly carried out on NVIDIA GTX 1080Ti GPU.

## 5.2 EVALUATION

**Baselines**  Besides the comparison against ground truth data, we also compare our model to two recent works that use data-driven approaches to simulate fluids under the Lagrangian framework. Ummenhofer et al. (2020) use the continuous convolutional kernel to learn fluid dynamics and they reported that their model has outperformed other works in this domain. Sanchez-Gonzalez et al. (2020) propose a graph network-based simulator (GNS) as a general-purpose physic simulator under Lagrangian representation. Our model and GNS both transform and pass messages of fluid field via graph structures, but GNS consists of a far larger and deeper network with multiple sub-blocks and thus contains much more parameters than ours. For all baseline models, we adopt the same training strategy from original papers but train them on our dataset (See A.5 for full detail).

**Metrics**  To conduct quantitative analysis, we evaluate model performance based on several metrics. We report the asymmetric version of Chamfer Distance between the simulated results of different models and ground truth sequence. The asymmetric Chamfer distance for two collections of particles $X, Y$ (from $X$ to $Y$) is defined as:

$$L(X, Y) = \frac{1}{N} \sum_{\mathbf{x} \in X} \min_{\mathbf{y} \in Y} d(\mathbf{x}, \mathbf{y}), \tag{19}$$

where $N$ is the total particle number of point cloud collection $X$, and distance function $d(\mathbf{x}, \mathbf{y})$ is evaluated using L2-norm $\|\mathbf{x} - \mathbf{y}\|_2$. We investigate two essential physical quantities for incompressible fluid simulation - velocity divergence and particle density deviation of fluid field. In addition, we use normalized mean absolute error (MAE) and relative tolerance to evaluate the error of advection net and pressure net on single frame inference respectively. The normalized MAE from prediction $\hat{y}$ to ground truth $y$ is defined as:

$$L_{\text{MAE}} = \frac{1}{N} \sum \frac{|\hat{y} - y|}{|y|}. \tag{20}$$

The relative tolerance of the numerical solution $\hat{\mathbf{x}}$ to a linear system $A\mathbf{x} = \mathbf{b}$ can be defined as:

$$tol = \frac{\|A\hat{\mathbf{x}} - \mathbf{b}\|_2}{\|\mathbf{b}\|_2}. \tag{21}$$

## 6 RESULTS

**Performance**  In order to measure the performance of our FGN model as a physic simulator, we performed simulations on several different test cases. In the first two cases, dam collapse and water

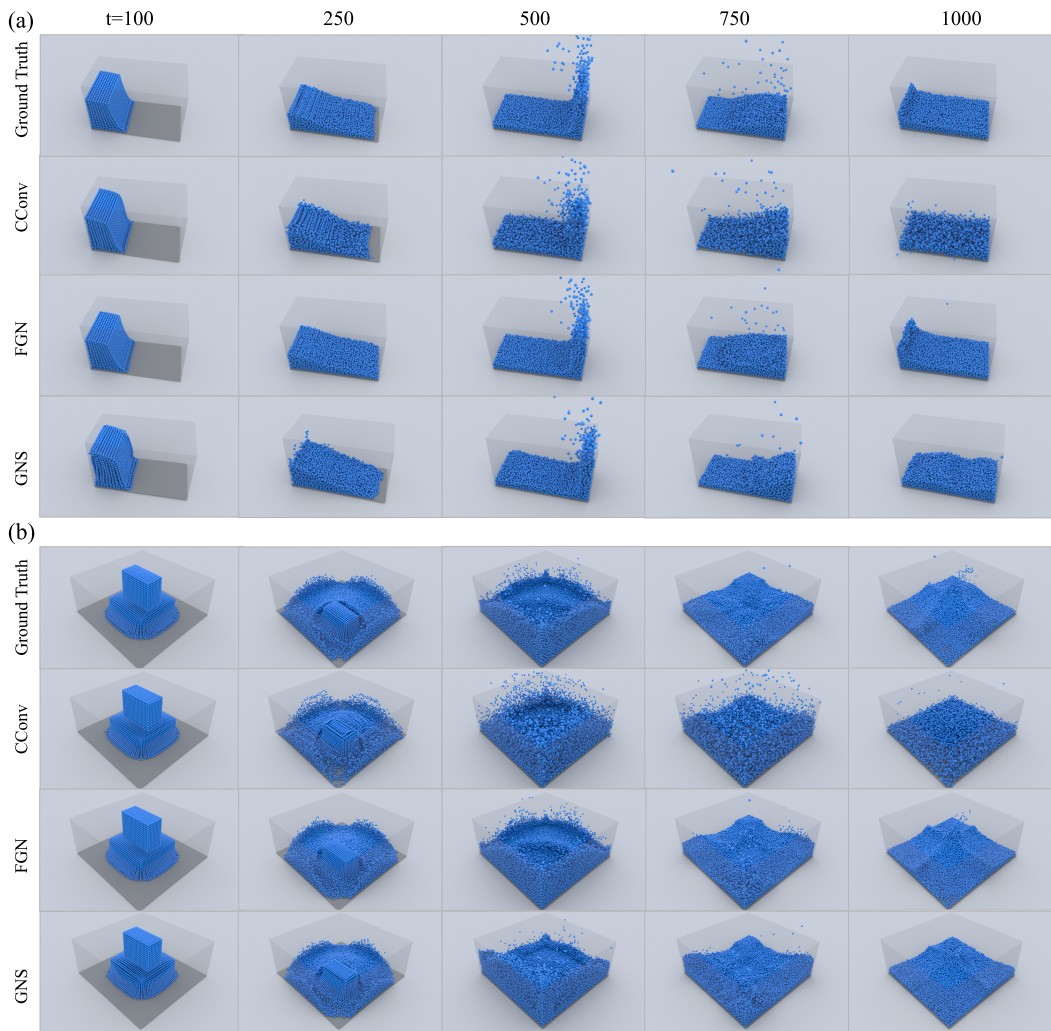

Figure 2: Qualitative analysis on: **(a)** Dam collapse. **(b)** Water fall. In CConv and GNS, we can observe oscillation on the free surface of fluids, while the results generated from our model maintains a much more compact and smoother shape. After long time steps, CConv's result fails to maintain smooth and compact fluid distributions and GNS' prediction is significantly slower than ground truth, while our model's prediction has minor difference from the ground truth.

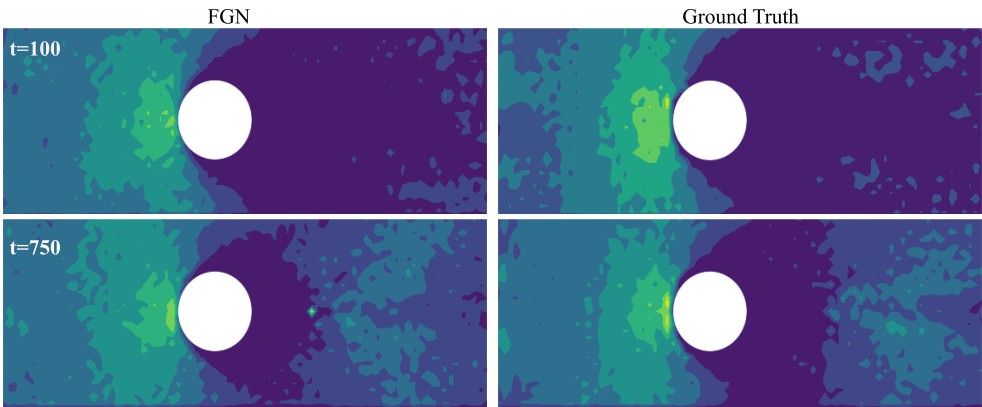

Figure 3: Pressure distribution contour. Our model can learn to generate reasonable pressure distribution. The distribution agreed with ground truth, which captures the shape of shifting high pressure and low pressure region.

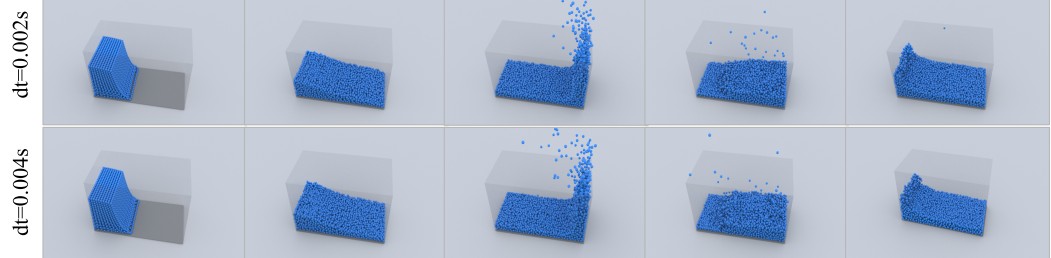

Figure 4: Qualitative comparison of model using different time step sizes. We use different time step size to simulate with our model. Despite the training set is generated with a time step size of 0.002s, our model can be generalized to a larger time step size with a neglectable increase in error. Other models diverge immediately when extrapolate to time step size different from training data.

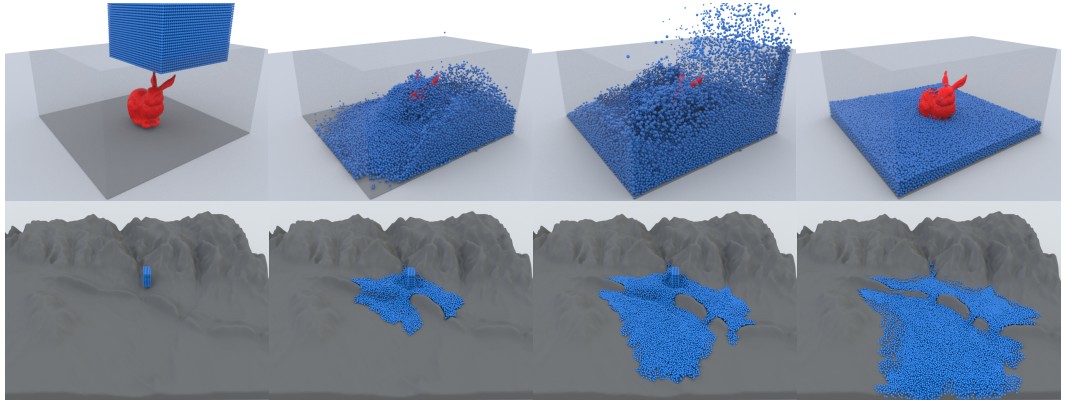

Figure 5: Generalization to complex geometries. Top: A fluid block drop on Stanford bunny. Bottom: A fluid emitter is placed at the front of Grand Canyon 3D model and emitted fluids gradually fill up reservoir at downstream. Although the training data only has basic geometries such as cube and cylinder, our FGN model can be generalized to complex geometries which are beyond the distribution of training data.

| Model | Dynamical system | Model Output | Evaluation metric | Error |
|---|---|---|---|---|
| Advection Net | $\dot{\mathbf{u}} = \mathbf{g} + \nu \nabla^2 \mathbf{u}$ | prediction of $\dot{\mathbf{u}}$ | Normalized MAE | $12.4\% \pm 5.6\%$ |
| Pressure Net | $\nabla^2 p = \frac{1}{\Delta t} \nabla \cdot \mathbf{u}^*$ | prediction of $p$ | Relative tolerance | $8.3\% \pm 1.1\%$ |

Table 2: Quantitative analysis of advection net and pressure net. We evaluate advection net's performance as solver for a forward problem. We generate ten sequences with different set of material parameters as test data for advection net. The gravity $\mathbf{g}$ for test data ranges from 1.0 to 100.0 and viscosity parameter $\nu$ ranges from 0.1 to 0.0001. For pressure net, we report its error as solver for a linear equation system. The test data for pressure net is generated using dam collapse, water fall, and Stanford bunny sequences.

fall, we qualitatively measure the results of different models via visualization[1] of fluids in Figure 4. We report the physical property of simulation results and its Chamfer distance to ground truth data. Quantitative results over the whole simulation sequence are listed in Table 1 (See Figure 9 in A.3 for error trend figures). The results show that our model FGN gives the best accuracy in retaining physical properties and position prediction. In addition, we study the performance of each sub-network in our model as stand-alone solvers for sub-dynamical systems and report their relative error in Table 2. For the advection net, we challenge it by applying a different set of material parameters (i.e. different gravity and viscosity parameters). We report the normalized mean absolute error (MAE) between prediction and ground truth. For pressure net, we evaluate the relative tolerance of its predicted solution $\hat{\mathbf{p}}$ to the discretized pressure poisson equation (i.e. $A\mathbf{p} = \mathbf{b}$). The relative low error demonstrates the capability of our sub-networks in learning and predicting physics.

---

[1]Video link: `https://sites.google.com/view/fluid-graph-network-video/home`

| Model | Parameters | Model inference time (ms) | NNS time (ms) |
|---|---|---|---|
| FGN (this work) | **41996** | **5.7** | 206.4 |
| GNS | 1406272 | 44.6 | 153.2 |
| CConv | 692902 | 28.9* | **24.7**[*] |
| MPS (ground truth) | - | 520.4 | 221.7 |

Table 3: Runtime analysis of different model. The test was carried out on a dam collapse scene containing approximately 40k fluid particles. We report the total trainable parameters for each deep learning model, averaged model inference time and nearest neighbor searching time. For ground truth solver, MPS, we report the time it used to calculate advection and pressure projection as model inference time. The pressure solver in MPS is a preconditioned conjugate gradient (PCG) solver implemented in Pytorch. Note that in CConv, the network and neighbor searching method are based on Tensorflow and Open3D (denoted with *). Besides CConv, all other model are implemented in Pytorch and use spatial hashing on GPU for neighbor searching. Our model has the smallest size and fastest inference time.

**Generalization**    To test out the model's capability of generalization. We apply our model to test cases with conditions that are beyond training distributions. In the first case, we study how our model will predict the pressure distribution of circular flow around a cylinder. This scene contains an inflow on the left side which keeps emitting particles during the simulation and an outflow on the right side. We challenge our model's robustness by applying different time step sizes to it. Figure 4 shows qualitative comparison of model output under different time step sizes. In addition, we simulate on two scenes which contain much more complex geometries. Visualizations of two test scenarios containing complex geometries are shown in Figure 5. Although our model is trained with only a fairly small dataset, it remains accurate under several different conditions. This demonstrates it capability of generalization and robustness. (More details on quantitative analysis are in A.3. Figure 10 shows position error trend under different time step sizes. Figure 11 shows position error trend under complex scenes.)

**Ablation Study**    We test the performance of different types of aggregators used in pressure net, as pressure net has the largest impact on overall prediction accuracy. [2] We compare our aggregator against graph convolution networks (GCN) from Kipf and Welling (2016), Hamilton et al. (2017)'s graph SAGE using mean aggregator , and MLP *w/o* any graph aggregation operation. In general, all aggregators give a similar performance on overall position prediction (similar Chamfer distance error), yet our model significantly improves the quality of predictions in terms of maintaining a constant density (See Table 4 in A.4 for full detail).

In addition, we report the model size (parameter number) and runtime benchmark in Table 3. Our model is very efficient in terms of training and inference, as it has far less trainable parameters than others.

# 7    CONCLUSION

In this paper, we present a data-driven Lagrangian fluid model for incompressible fluid simulation by decomposing simulation scheme as separate reasoning tasks based on Navier-Stokes equation. It can preserve many essential physical properties of the fluid field such as low volume compression, and predict reasonable pressure distribution. Our model also has generalization capability, where it can remain stable when extrapolating to a wide range of different geometries and adopting different time step sizes. In general, our work is an advance in learning on unstructured data with graph neural networks, and enriches the paradigm of combining learning-based methods with physical models as well.

---

[2]During the preliminary experiment, we find that aggregators in advection net do not have significant impact on overall model performance when viscosity parameter of fluids is small, therefore ablation results on it are not listed here; Aggregator in collision net is just summing up all the predicted edge attributes and conducting linear transformation. Using more advanced structure does not bring in further improvement.

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

APPENDIX A

A.1 IMPLEMENTATION DETAILS

**Graph construction** We build the graph by establishing edges between particles within limit radius. We perform the neighborhood searching on GPU by using cell sort algorithm. All the edge attributes are stored in sparse matrices. Although the limit radius does not have significant impact on training loss and larger limit radius will increase the computing cost drastically, we found that small limit radius can influence the long-term stability of the model. For advection net and pressure projection net, we select the limit radius to be three times the particle diameter D (D = 0.050m), and 0.9D for collision net.

**Numerical model** In general, we calculate all the gradient operators and smooth kernel based on numerical model of SPH.

In SPH model, particle density is defined as:

$$n_i = \sum_{i \neq j} W\left(\left|\mathbf{r}_i - \mathbf{r}_j\right|, h\right), \forall j \in \mathcal{N}(i). \tag{22}$$

For scalar quantity $\phi_i$ at location $\mathbf{r}_i$, we approximate its gradient by:

$$\nabla \phi_i = \frac{d}{n^*} \sum_j (\phi_j - \phi_i) \nabla W_{ij}, \tag{23}$$

where $n^*$ is the constant particle number density derived by calculating the maximum particle number density at the initial frame, $d$ is the dimension of the problem, $W_{ij}$ denotes the smooth kernel function value between particle $i$ and $j$. Similarly velocity divergence is defined as:

$$\nabla \cdot \mathbf{v}_i = \frac{d}{n^*} \sum_j (\mathbf{v}_j - \mathbf{v}_i) \cdot \nabla W_{ij}, \tag{24}$$

We adopt the same smooth kernel function from Koshizuka and Oka (1996), which is very simple to evaluate.

$$W_{ij} = \begin{cases} \frac{h}{\|\mathbf{r}_i - \mathbf{r}_j\|_2} - 1 & \text{if } \|\mathbf{r}_i - \mathbf{r}_j\|_2 < h \\ 0 & \text{if } \|\mathbf{r}_i - \mathbf{r}_j\|_2 \geq h \end{cases} \tag{25}$$

A.2 TRAINING

**Dataset Generation** We place one or two of the following basic obstacles (as shown in Figure 6) in training scenes. In addition to obstacles, each scene is a cubic box (80x80x40) containing a fluid block (25x25x10) (as shown in Figure 7). We place the fluid block at random place in the box and initialize its velocity of one direction by uniformly sampling from $\mathcal{U}(0, 0.1)$. We generated 20 scenes adopting above settings and simulate each training scene to 1000 time steps with step size $dt = 0.002$.

**Strategy** In each time step of the ground truth simulator, there are mainly three steps. First, advect fluids with body force and viscosity:

$$\mathbf{v}^* = \mathbf{v}^n + \mathbf{a}^{\text{adv}} \Delta t, \tag{26}$$
$$\mathbf{x}^* = \mathbf{x}^n + \mathbf{v}^* \Delta t, \tag{27}$$

and then solve the pressure poisson equation,

$$\nabla^2 p = \frac{1}{\Delta t} \nabla \cdot \mathbf{v}^*, \tag{28}$$

lastly,

$$\mathbf{v}^{n+1} = \mathbf{v}^* - \frac{\nabla p}{\rho_c} \Delta t, \tag{29}$$
$$\mathbf{x}^{n+1} = \mathbf{x}^* + \mathbf{v}^{n+1} \Delta t. \tag{30}$$

Here, we use $[\mathbf{x}^*, \mathbf{v}^*, \mathbf{g}, \nu]$ as the training features and $\mathbf{a}^*$ (i.e. $(\mathbf{v}^* - \mathbf{v}^n)/\Delta t$) as the training label for advection net. $[\mathbf{x}^*, \mathbf{v}^*]$ and particle density $\rho$ (evaluated based on $\mathbf{x}^*$) are the training features for pressure net, with pressure $p$ as label. To train collision net, we simulate another particle system that is updated only based on elastic collision rule and applies no other dynamics. We use relative velocity and position before collision as inputs, and use velocity difference (i.e. $\Delta v$) as output target.

We train each network for 100,000 iterations of gradient updates and decay learning rate from 0.001 to 0.0000625. For the training of three sub-networks (advection net, collision net, pressure net), the batch size of each network is 16, 4 and 32 respectively. To allow the mini-batching of different graph, we mini-batch the adjacency matrix of different scenes by creating a large sparse matrix and stacking adjacency matrix on the diagonals.

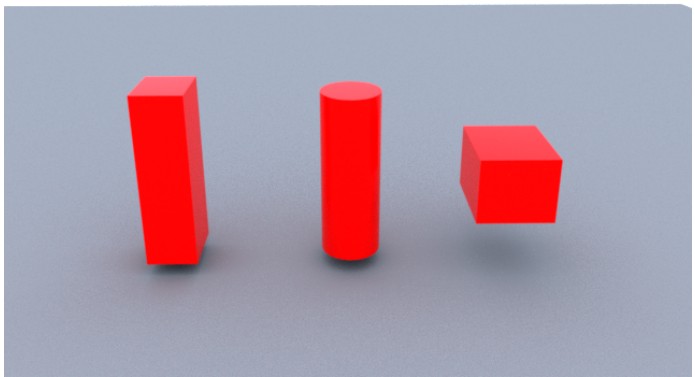

Figure 6: Three kinds of obstacles: Square pillar(width:10, length:10, height:40), cylinder(radius:4, height:40) and cube(15x15x15)

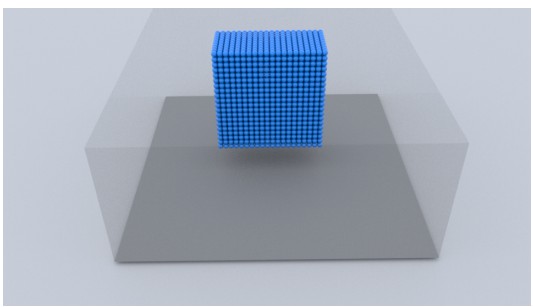

Figure 7: Fluid block and box

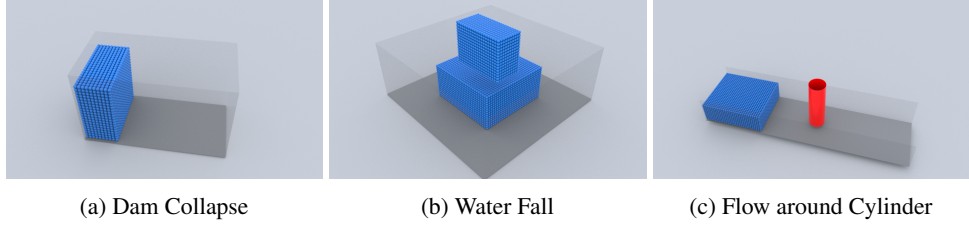

(a) Dam Collapse        (b) Water Fall        (c) Flow around Cylinder

Figure 8: Visualizations of initial condition settings in evaluation scenarios

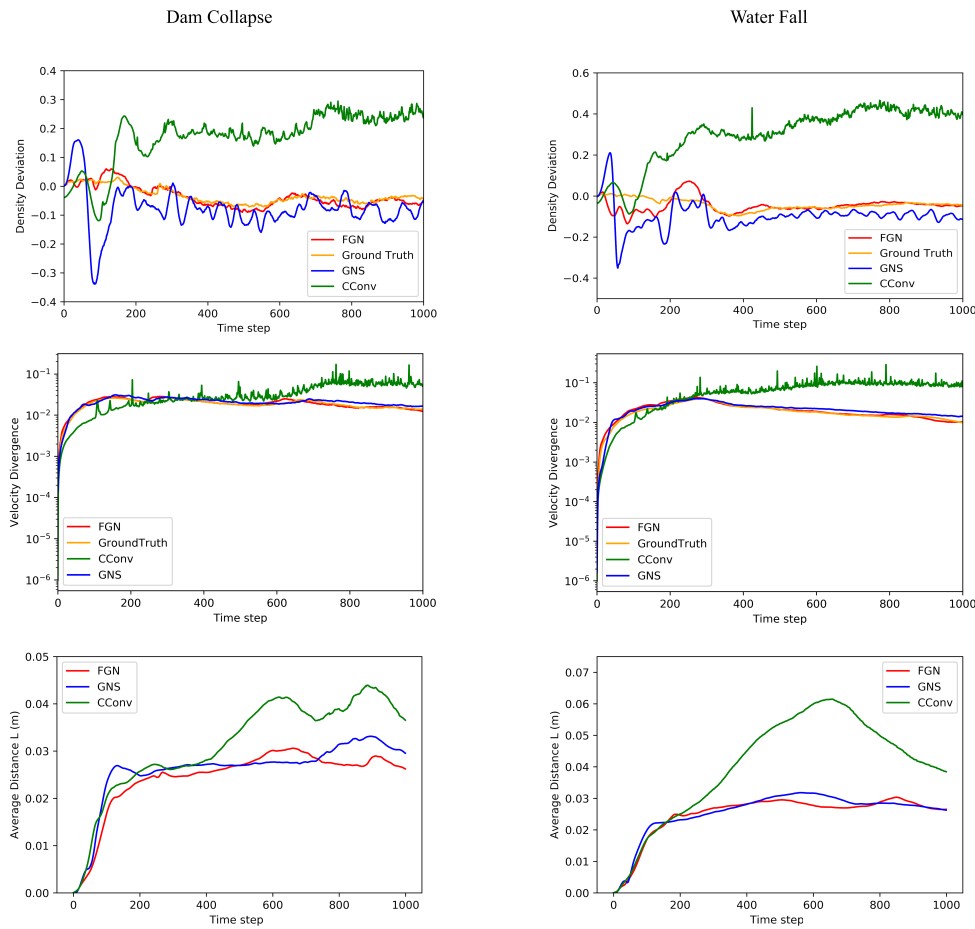

Figure 9: Error trend with respect to time. The density deviation of our model remain close to zero despite some oscillation at first. Although GNS also maintains a low density deviation, its average density oscillates more severely. This is consistent with the qualitative comparison, in which the fluid surface from GNS is oscillating and less compact. CConv fails to maintain constant density and its density increases significantly after collision to the wall boundary. Both our model and GNS can predict a low divergence velocity field while CConv fails to capture this property. This somehow explains why CConv struggles to maintain constant density.The average Chamfer distance to the ground truth data accumulates at first and stabilizes after system is in equilibrium. CConv has larger average Chamfer distance while graph-based model's distances are smaller.

## A.3 METRICS EVALUATION DETAILS

We show the error trend of dam collapse and water fall scene under different evaluation metrics in Figure 9. The position loss analysis of different time step sizes is shown in Figure 10. Position error of complex scenarios is shown in Figure 11.

## A.4 ABLATION STUDY DETAILS

Performance of different graph message aggregate structures is listed in Table 4.

## A.5 BASELINES IMPLEMENTATION

**Continuous Convolution**   We use the open-source implementation from Ummenhofer et al. (2020) [3]. To give a fair benchmark result we train their network with our dataset. Note our training dataset is much smaller than theirs and does not include complex geometries. As in our work, we model

---

[3]https://github.com/intel-isl/DeepLagrangianFluids

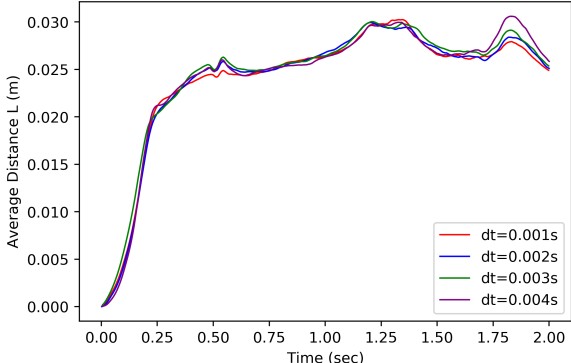

Figure 10: Average Chamfer distance to the ground truth data of FGN model under different time step sizes. FGN can generalized to different time scale with minor difference in performance. The training data uses a time step size of $dt = 0.002s$, and FGN converges for all $dt < 0.005s$.

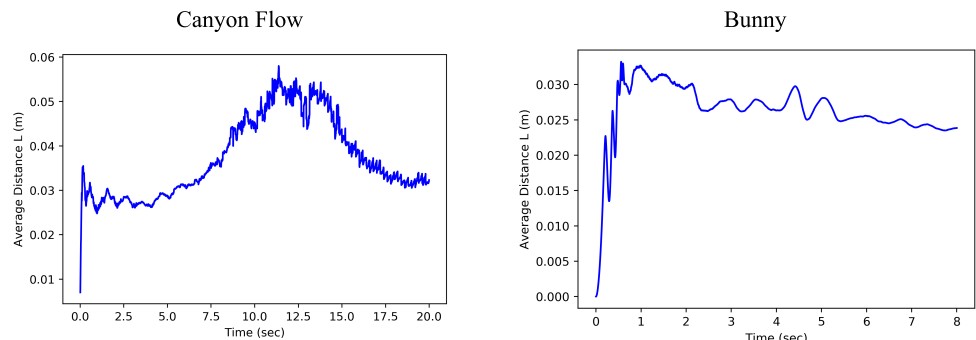

Figure 11: Average Chamfer distance to the ground truth data of complex scenes. For both scenes, at first, position error accumulates quickly as system evolves. At some point, the error starts to decrease until reaching an equilibrium.

| Case | Model | Density error | | Velocity divergence | | Average Chamfer distance (mm) | |
|------|-------|------|------|------|------|------|------|
| | | Mean | Max | Mean | Max | Mean | Max |
| Dam Collapse | FGN (this work) | **0.0461** | **0.0900** | **0.0195** | **0.0280** | **24.2** | **30.1** |
| | GCN | 0.0939 | 0.3832 | 0.0196 | 0.0293 | 27.7 | 31.6 |
| | SAGE | 0.0935 | 0.3597 | 0.0196 | 0.0289 | 26.4 | 31.3 |
| | MLP | 0.0963 | 0.3624 | 0.0196 | 0.0289 | 28.5 | 32.8 |
| | Ground Truth | 0.0380 | 0.0710 | 0.0190 | 0.0268 | - | - |
| Water Fall | FGN (this work) | **0.0541** | **0.1350** | **0.0207** | **0.0431** | **24.8** | **29.6** |
| | GCN | 0.0794 | 0.3373 | 0.0210 | 0.0433 | 25.5 | 31.7 |
| | SAGE | 0.0769 | 0.3254 | 0.0209 | 0.0432 | 25.3 | 31.4 |
| | MLP | 0.0830 | 0.2967 | 0.0209 | 0.0433 | 26.3 | 32.4 |
| | Ground Truth | 0.0429 | 0.0966 | 0.0196 | 0.0398 | - | - |

Table 4: Quantitative ablation study. We compare our aggregator against graph convolution networks (GCN) from Kipf and Welling (2016), Hamilton et al. (2017)'s graph SAGE using mean aggregator and MLP *w/o* any graph aggregation operation. All aggregators have two layers.

solid obstacles and wall as virtual particles, so we transform these virtual particles into surface and corresponding normals before inputting them into CConv network. The original CConv uses a time step size of 0.02s, but given such a large time step size, the qualitative results can be distinct from ground truth and other model with smaller time step size. Hence during training and comparison we adopt a time step size of 0.002s for CConv model.

**Graph Network-based Simulator**   We implemented GNS following the description in Sanchez-Gonzalez et al. (2020). We build GNS with 10 unshared GN blocks, conditioned on 5 previous velocities and input relative positions as edge features. We chose the connectivity radius to be 2.1D, so that the number of neighbors is around 20. Sanchez-Gonzalez et al. (2020) use finite difference to calculate acceleration and velocity, but in our implementation we explicitly maintain an array to store the velocities of all particles. Additionally, we do not use learned embedding but simple zero and one to indicate particle material type, as in our testing domain there are only two kinds of particles - solid and fluid. The loss function and training procedure are implemented as described in Sanchez-Gonzalez et al. (2020), including noise injection and similar normalization techniques.

For the implementation of GN block, as Sanchez-Gonzalez et al. (2020) states "We use GNs without global features or global updates (similar to an interaction network)", so we implement the GN block update mechanism following the description in Battaglia et al. (2018).

$$\mathbf{e}'_k = \phi^e(\mathbf{e}_k, \mathbf{v}_{r_k}, \mathbf{v}_{s_k}, \mathbf{u}) \tag{31}$$

$$\mathbf{v}'_i = \phi^v(\hat{\mathbf{e}}'_k, \mathbf{v}_i, \mathbf{u}) \tag{32}$$

$$\hat{\mathbf{e}}'_i = \rho^{e \to v}(E'_i) \tag{33}$$

where $\phi$ is the update function and implemented as MLP here, $\rho$ is aggregation function which aggregates all the edge attributes to its center vertex, $\mathbf{u}$ is the global feature and here we append them to the nodal features as input feature, $r_k$ denotes receiver vertices and $s_k$ denotes sender vertices.

In the testing stage, as authors did not state how to set initial velocities, so we just warm start the simulation by calculating the first 5 frames using MPS method and apply GNS to the rest frames.

