# OpenReview forum: "Learning Lagrangian Fluid Dynamics with Graph Neural Networks"
_ICLR.cc/2021/Conference — Reject_

### Official Review · AnonReviewer3 · 2020-10-25
**Potentially interesting results, needs some revision**

**Rating:** 4
**Confidence:** 4

**Review:**

The paper deals with the prediction of 3D Lagrangian Fluid Simulations. Therefore the problem is divided into 3 subproblems, oriented on numerical simulations. An advection part, where the acceleration of the particles is calculated, a collision step, where the boundary effects are included, and a pressure prediction part, where the pressure for maintaining the volume is determined. A graph-based network is used for each part, which is either node or edge-based according to the requirements.

I think it's a good idea to divide the problem into single subtasks and to orientate on the numerical method. I also find the evaluation of physical properties like density and velocity divergence very relevant and show in my eyes the superiority of the method compared to the state-of-the-art methods. However, there are still some open questions:
* It is mentioned that tests with different time steps were made as proof of generalization, but I cannot see any corresponding results.
* Also, I'm not quite sure what the advection network and the collision network calculate exactly. I think it would be more logical if the advection network also performs the integration step, so you could approximate higher-order integration schemes. With the collision network, I'm not quite sure what the GT should be. In most SPH solvers the boundaries are included directly in the pressure computation and not in a separate collision step. In general, I would like to have an ablation study where the relevance of the individual components is more obvious.
* Another thing is that force-based SPH methods, as adapted in this paper, are usually less stable than position-based methods (PBF), from which state-of-the-art methods are based. I would therefore be interested in a comparison in an extreme scenario, e.g. a setup with a very high water column. I could imagine that the method would fail.
* Finally, I would recommend adding some videos to the supplementary material, which could prove the temporal coherence of the method.

There are a few things I noticed about the text itself:
* First, there are some linguistic errors in the text.
* Furthermore I would recommend naming the intermediate results in the equations (e.g. equation 10) differently. This can be a bit confusing.
* As of the last point I found the section about network architecture a bit short and not very informative. I would make it a bit longer and more detailed.

All in all, I find the method interesting, but in my eyes, there are still some missing points. Unfortunately, I think that the text needs a revision in general because there are quite a lot of linguistic errors. Therefore I would rather vote for a reject.

---

> ### Author Response · Authors · 2020-11-20
> **Responses to Reviewer4**
>
> We thank the reviewer for their time and helpful feedback.
>
>
> Q:\
> “It is mentioned that tests with different time steps were made as proof of generalization, but I cannot see any corresponding results.”
>
> A:\
> In the original version of the paper, we have a qualitative comparison of model results using dt=0.002s/0.004s in Figure 3. In the revised version, this figure is moved to Figure 4. We also added the figure of position error trend under different time step sizes in the Appendix (Figure 10).
>
>
>
> Q:\
> “Also, I'm not quite sure what the advection network and the collision network calculate exactly. I think it would be more logical if the advection network also performs the integration step, so you could approximate higher-order integration schemes. With the collision network, I'm not quite sure what the GT should be. In most SPH solvers the boundaries are included directly in the pressure computation and not in a separate collision step. In general, I would like to have an ablation study where the relevance of the individual components is more obvious.”
>
> “Another thing is that force-based SPH methods, as adapted in this paper, are usually less stable than position-based methods (PBF), from which state-of-the-art methods are based. I would therefore be interested in a comparison in an extreme scenario, e.g. a setup with a very high water column. I could imagine that the method would fail.”
>
> A:\
> Yes, the collision net is not necessary for boundary handling, just like most force-based methods, the pressure net is responsible for imposing boundary conditions.  It is also true that force-based methods are less stable in extreme cases (e.g. fluids initialized with relatively high velocity), we have observed this in both ground truth solver and our model. This is why we add a collision net, a network that emulates elastic collision, in order to alleviate particle penetration in many extreme cases.
>
> Advection net is calculating a sub-dynamical system that only contains body force and viscosity:
> a^{adv} = g + nu \nabla^2 v, and indeed it will integrate forward fluids to the intermediate state.
>
> Collision net is predicting the elastic collision effect, we train it using a particle system that is updated only based on the elastic collision rule.
>
>
>
> Q:\
> “Finally, I would recommend adding some videos to the supplementary material, which could prove the temporal coherence of the method.”
>
> A:\
> Thanks for the suggestion.
> We have added videos for all test sequences, which can be found in:
> https://sites.google.com/view/fluid-graph-network-video/home
>
>
>
> Q:\
> “Furthermore, I would recommend naming the intermediate results in the equations (e.g. equation 10) differently. This can be a bit confusing.”
>
> A:\
> We renamed the intermediate results (previous: v^* -> now:v^**) after passing through the collision net.

---

### Official Review · AnonReviewer1 · 2020-10-26
**Recommend to reject**

**Rating:** 4
**Confidence:** 4

**Review:**

### Summary
The paper presents a method for learning Lagrangian fluid dynamics from MPS data. By injecting domain knowledge, and separately training subcomponents of the solver, it achieves low error rates on e.g. divergence.

### Recommendation
There are a lot of different approaches for learning physical dynamics, which differ in the amount and type of inductive biases and domain knowledge injected. This is often a tradeoff between generality versus accuracy, and the question of what is a useful balance that has actual advantages over classical solvers is still open.

That said, I find it hard to make the case for the balance chosen in this paper; the architecture is limited to data from a specific solver (MPS) so unlike general end-to-end approaches (e.g. GNS [Sanchez et al 2020], CConv [Ummenhoffer et al 2019]) it's not a pathway to learning from general data. On the other hand, unlike methods that use a lot of domain knowledge to be more efficient than the ground truth solver (e.g. [Ladicky et al.]), this paper doesn't demonstrate any concrete advantages over MPS. It also doesn't study the impact of its individual choices in great detail, and does not back up some of its claims (generality & generalization, see detailed comments). In its current form, I'm not sure we will learn very much from this method, and hence I recommend rejecting this paper.

### Detailed comments
- Model: The architecture is very close to the SPH/MPS algorithm in structure. The main difference to a hard-coded solver seems to be that the interpolation kernel is learned, and there is a processor MLP after each block which can act as a corrector. In between sub-block, the output is projected back to position, velocity, so the NN can't learn a richer intermediate representation. The biggest limitation I see it caused by supervising the 3 subcomponent separately. If I understand this correctly, this means like you a) can only train on a solver which has exactly these subcomponents (i.e. MPS, even SPH works slightly differently), b) you need access to solver internals, and c) you will just learn to replicate the solver internals-- and if you can't learn e.g. a better kernel than the solver, why learn it at all?
- Claims:
  1. Abstract: *"our model can adopt a range of time step sizes different from ones using in the training set"*. While different step sizes are *mentioned* in the results, there is no numbers, figures or other details to substantiate this claim.
  2. Conclusion: *"Our model also has generalization capability, where it can remain stable when extrapolating to a wide range of different geometries"*. The only result related to this I could find is fig. 4. The dataset description was a bit vague, so it's a bit unclear to me that this really is an extrapolation test. Was the cylinder among the objects in the training set? In any case, I don't think a test on a single cylinder obstacle support the statement of "a wide range of different geometries".
  3. Intro: *"...it can be extended to simulation of other dynamics under Lagrangian framework"*. I'm not sure this is the case; all the specific details which make this method different from other graph-based approaches *are* specialized to fluids, specifically MPS. How would you e.g. extend this method for elastics or granular materials? For those, there is no pressure or density (depending on approach), but strain/stress etc., and it's unclear which sub-module you'd need or how to supervise them. I'm not even sure this approach would work for Lagrangian fluid data from a different solver (e.g. MPM).
  4. Although not a direct claim, the method is motivated by the computational cost and bad scaling behavior of classical solvers. However, is this method actually faster than the GT simulator, or can demonstrate better scaling? There's no results on this, and since the method is directly supervised on individual solver components it'd actually be surprised if that is the case.
- Results: The main result seems to be that divergence is better preserved than in end-to-end approaches like GNS and CConv. This finding is not very surprising, considering that this method directly supervises on the pressure correction module, while the baselines do not. I find it more surprising that the difference to GNS is rather small in fig. 6, even though it does not have a notion of pressure at all.

### Additional questions
- Just to make sure I have understood the method correctly, how exactly to you train the 3 modules? I.e. where do the labels for supervision come from, are those taken from the corresponding modules in the ground truth simulator?
- Where does the density come from? Is this computed with a hard-coded interpolation kernel like in SPH? And if yes, why-- as the other modules are specifically build around learning a kernel?
- Where do you see the advantages of your method compared to using the ground-truth solver?

### Further comments to the authors
While I recommended rejecting this paper in its current form, I think that this is an exciting area to perform research in, and there are many directions worth exploring. I think it's worth thinking about specifically what advantages a learned or semi-learned method can have over ground-truth solvers, and then finding ways of capitalizing on them. Some papers have explored performance, or generality, and the elephant in the room is how to learn from actual real-world sensor data while still exploiting domain knowledge. Also, there's a multitude of papers pushing a specific method, but very little work on thoroughly studying the concrete impact of all the little tricks, choices in representation, and the amount and concrete form of domain knowledge injected.

---
Update: The added generalization tests and performance numbers strengthen the paper, and make the aim more clear. The paper seems now mostly focused on accelerating MPS (or potentially other SPH variants). While the new result table shows speedups compared to MPS for model inference, in practice the model still is bottlenecked by NN search, and total runtime of inference+NN is not significantly different to MPS. And if speed is the aim, the appropriate baseline comparisons should be to methods which -- as this method-- use domain knowledge and solver internals to speed up the solver (e.g. Ladicky et al.).
On the method side, it does seems like the proposed approach is limited by supervising specific MPS subcomponent separately, which hinders learning richer intermediate representations, applying the model to other systems, or taking significantly larger timesteps. There could be potential benefits of the specific architecture chosen, but their effect is a bit unclear; as for the comparisons shown the most significant effect is supervision on subcomponents.
I will therefore keep my score.

---

> ### Author Response · Authors · 2020-11-20
> **Responses to Reviewer 3 (Part1)**
>
> We thank the reviewer for their time and detailed comments.
>
>
> Q:\
> “On the other hand, unlike methods that use a lot of domain knowledge to be more efficient than the ground truth solver (e.g. [Ladicky et al.]), this paper doesn't demonstrate any concrete advantages over MPS. It also doesn't study the impact of its individual choices in great detail, and does not back up some of its claims (generality & generalization, see detailed comments). ”
>
>
> A:\
> We have now added model size and runtime analysis in Table 3, and added more details on how we generate training data in the Appendix (A.2 Dataset generation). We also added two more tests on more complex geometries (Stanford bunny and Grand Canyon) and error evaluation figure of model under different time step sizes in the Appendix (Figure 10).
>
> The biggest advantage of using physics prior knowledge to design the model is that it significantly improves the calculation efficiency. As shown in the model size and runtime analysis, our model is quite small (<50k parameters in total) and efficient in inference.  In addition, the dataset we use is simple and small. It has 20 trajectories in total and only includes simple geometries like cubic block, cylinder, square pillar. However, our model can be generalized to larger and more complex scenes. This demonstrates that by using domain knowledge the training process can also be simplified.
>
>
> Q:\
> “The biggest limitation I see it caused by supervising the 3 subcomponent separately. If I understand this correctly, this means like you a) can only train on a solver which has exactly these subcomponents (i.e. MPS, even SPH works slightly differently), b) you need access to solver internals, and c) you will just learn to replicate the solver internals-- and if you can't learn e.g. a better kernel than the solver, why learn it at all?”
>
>
> A:\
> The main reason we choose to design the model as three sub-networks (especially advection net and pressure net), is based on the observation that many force-based SPH (MPS and other similar SPH variants) adopt a similar scheme: advection based on body force and viscosity -> pressure projection (or other similar forces that emulates pressure). Using this domain knowledge, we also decompose the simulation scheme in our model to make the model easier to train. As shown in Table 3 of the revised paper, the main problem of MPS (and many other SPH methods that prioritize accuracy) is its high computational cost. Therefore we propose a GNN model to learn a fast approximation of this process without too much loss in accuracy.
>
> As for the collision net, it can actually be trained on any particle system. We use a particle system that is only updated based on elastic collision rule and contains no other dynamics to generate training data for collision net.
>
> Another advantage of designing the advection net as a separate model is that the model can be more easily extended to fluids with different material parameters (g and nu) using only a fairly small training dataset.
>
>
> Q:\
> “Abstract: "our model can adopt a range of time step sizes different from ones using in the training set". While different step sizes are mentioned in the results, there is no numbers, figures or other details to substantiate this claim.”
>
>
> A:\
> In the original version of the paper, we have a qualitative comparison of model results using dt=0.002s/0.004s in Figure 3. In the revised version, this figure is moved to Figure 4. We also added the position error trend figure in the Appendix (Figure 10).
>
>
> Q:\
> “Conclusion: "Our model also has generalization capability, where it can remain stable when extrapolating to a wide range of different geometries". The only result related to this I could find is fig. 4. The dataset description was a bit vague, so it's a bit unclear to me that this really is an extrapolation test. Was the cylinder among the objects in the training set? In any case, I don't think a test on a single-cylinder obstacle support the statement of "a wide range of different geometries".”
>
>
> A:\
> Thanks for pointing this out.
> We have added tests on two more complex scenes and more detail on training data distribution can be now found in the Appendix. Dataset generation (Figure 6 and Figure 7).

---

> > ### Author Response · Authors · 2020-11-20
> > **Responses to Reviewer 3 (Part2)**
> >
> > Q:\
> > “Intro: "...it can be extended to simulation of other dynamics under Lagrangian framework". I'm not sure this is the case; all the specific details which make this method different from other graph-based approaches are specialized to fluids, specifically MPS. How would you e.g. extend this method for elastics or granular materials? For those, there is no pressure or density (depending on approach), but strain/stress etc., and it's unclear which sub-module you'd need or how to supervise them. I'm not even sure this approach would work for Lagrangian fluid data from a different solver (e.g. MPM).”
> >
> >
> > A:\
> > In our model, advection net and pressure net are performing very different tasks, but their networks’ architectures are exactly the same (with only different input and output). This somewhat demonstrates node-focused network’s capability to be extended to predict physical quantities of particles under a different dynamical system.
> >
> >
> > Q:\
> > “Although not a direct claim, the method is motivated by the computational cost and bad scaling behavior of classical solvers. However, is this method actually faster than the GT simulator, or can demonstrate better scaling? There's no results on this, and since the method is directly supervised on individual solver components it'd actually be surprised if that is the case.”
> >
> >
> > A:\
> > Thanks for pointing this out. We have added runtime analysis in Table 3. Our model is much faster in frame inference compared against baselines and ground truth solver.
> >
> >
> > Q:\
> > “Just to make sure I have understood the method correctly, how exactly to you train the 3 modules? I.e. where do the labels for supervision come from, are those taken from the corresponding modules in the ground truth simulator?”
> >
> >
> > A:\
> > We added a detailed description of the training strategy in the Appendix. A.2 Strategy.
> >
> >
> > Q:\
> > “Where does the density come from? Is this computed with a hard-coded interpolation kernel like in SPH? And if yes, why-- as the other modules are specifically built around learning a kernel?”
> >
> > A:\
> > We approximate density using a hard-coded interpolation kernel like in SPH. \
> > For pressure net, the learned kernel produces a fast and relatively accurate approximation of pressure. Hence we do not need any iterative solver (e.g. CG, PCG) to solve pressure now.
> >
> >
> > Q:\
> > “Where do you see the advantages of your method compared to using the ground-truth solver?”
> >
> >
> > A:\
> > We think the biggest advantage of our model is its calculation efficiency. First, it can be trained using a fairly small dataset. Second, it contains only a small amount of parameters and thus it is very fast in inference.

---

### Official Review · AnonReviewer4 · 2020-10-27

**Rating:** 5
**Confidence:** 4

**Review:**

Summary:
This paper presents a graph neural network approach specialized for Lagrangian fluid simulation.
Instead of an end-to-end approach a multi-step solution is proposed which divides the computations into 3 distinct networks.
The structure of the intermediate results computed by the networks follows the structure of solvers like PBF.
A goal of this approach is to more accurately model physical properties of fluids.
The experiments show comparisons with recent state-of-the-art methods and compare different information aggregation schemes.


Score:
Although I think the idea is interesting I tend towards reject.
The reason for that is mainly the evaluation, which uses very limited data (number of scenes, sequence length, variety) and is not very convincing.
This can be fixed by increasing the test set size and even test on datasets from other works with more variety.
Another reason is the very specific network design with multiple small parts in combination with the small timesteps.
In comparison with a single end-to-end trained network this approach is less practicable.
Additional experiments can help to strengthen the design choices that were made.


Strengths:
* The paper examines important physical quantities such as the density error and velocity divergence.
* The method is parameter efficient and works with different time steps without retraining.
* The description of the implementation is easy to follow and Figure 1 gives a good overview.


Weaknesses:
* The proposed network structure is very specific to the problem and the solver used for generating the ground truth.
  Each of the 3 networks is trained separately imposing the need to compute the intermediate quantities for each of the networks.
  The sequential network design allows to factor out quantities like the time step which helps with generalization but the strong supervision and the sequential structure also significantly limits the power of the network.
  This aspect needs to be investigated in detail as this distinguishes this work from other approaches which train end-to-end.
  It is possible to backpropagate through the update scheme (equations 7 to 12) during training and compare this to the proposed solution.
  Another comparison can be to remove the intermediate results and let the network output all desired physical quantities in the last layer.
* The evaluation (Table 1,2) seems to be based on only 2 sequences, which is not enough.
  The description of the dataset generation process is very short and important information is missing.
  Are there different types of obstacles or only cylinders? Does the number of particles differ for each scene? What is the number of fluid blocks per scene?
  Without knowing the variance of the training data and a small test set it is not clear if the evaluation is meaningful.
* The sequences are very short. The sequence length is only 2 seconds.
  This is not enough time for the fluid to reach a steady-state, thus the behavior of the model for this particular but important state is not well studied.
* The generalization to different time steps is good but the generalization to different scenes seems very limited.
  All scenes are box-like environments.
* Learning physics is often motivated by reducing computation costs but there is no information about the computation time.
  Further, the time steps used in the paper are quite small, which increases the number of iterations needed for long simulations and significantly simplifies the learning task.


Questions:

Section 5.1 mentions that the particle density inputs are normalized for the pressure network.
This is counterintuitive since density is an important feature for computing the pressure.
Why is this necessary to stabilize the training?



Minor comments:

There are some minor problems with the writing. Here are some for the first page:

Abstract:
- Our model uses _a_ graph ...

Introduction:
- fluids is an essential -> fluids _are_
- a large class of numerical models have -> a large class of numerical models _has_
- usually increase drastically when resolution -> usually _increases_ drastically when _the_ resolution
- data-driven model -> data-driven _models_
- dynamics under Lagrangian -> dynamics under _the_ Lagrangian
- universal feature (...) under Lagrangian framework as input -> universal _features_ (...) as input

Related Works:
- build upon Lagrangian representation of fluid -> build upon a Lagrangian fluid representation
- when material interface -> when the material interface

Will the code be released? If not it would be important to have a section in the appendix with all the important parameters for reproducing the networks in tabular form (layer sizes, normalizations, activations, etc.).

---

> ### Author Response · Authors · 2020-11-20
> **Responses to Reviewer2**
>
> We thank the reviewer for their time and detailed comments.
>
>
> Q:\
> “The sequential network design allows to factor out quantities like the time step which helps with generalization but the strong supervision and the sequential structure also significantly limits the power of the network. This aspect needs to be investigated in detail as this distinguishes this work from other approaches which train end-to-end.”
>
>
> A:\
> The strong supervision and the sequential structure do limit the model to be trained in an end-to-end way at this stage.  However, strong supervision and sequential design allow a significant reduction in model parameters, and our model requires only a small amount of training data. (We have added inference time benchmark and model size comparison in Table 2, also more details for training data generation are added to Appendix. Dataset Generation)
>
>
> Q:\
> “The evaluation (Table 1,2) seems to be based on only 2 sequences, which is not enough. The description of the dataset generation process is very short and important information is missing. Are there different types of obstacles or only cylinders? Does the number of particles differ for each scene? What is the number of fluid blocks per scene? Without knowing the variance of the training data and a small test set it is not clear if the evaluation is meaningful.
> The sequences are very short. The sequence length is only 2 seconds. This is not enough time for the fluid to reach a steady-state, thus the behavior of the model for this particular but important state is not well studied.”
>
>
> A:\
> Thanks for pointing out.
> We have now extended the description of data generation in the Appendix, including fluid block shape, solid obstacles’ shapes and amount. We also extended the test to two more complex scenes, and provide their error figures in the Appendix (Figure 10 and 11). In addition, we added videos of test scenes that contain more time steps.
> See: https://sites.google.com/view/fluid-graph-network-video/home
>
>
> Q:\
> “The generalization to different time steps is good but the generalization to different scenes seems very limited. All scenes are box-like environments.”
>
>
> A:\
> We added a test that simulates a large open scene with rather complex geometries (a Canyon 3D model) using our model. Screenshots are in Figure 5 and the video can be found on the above website.
>
>
> Q:\
> “Learning physics is often motivated by reducing computation costs but there is no information about the computation time. Further, the time steps used in the paper are quite small, which increases the number of iterations needed for long simulations and significantly simplifies the learning task.”
>
>
> A:\
> We have added inference time benchmark and model size comparison in Table 2.
> Small step size indeed decreases the difficulty for inference on single frames, but we also observed that as more time steps are involved, accumulated position errors are higher for end-to-end learning models (especially for CConv, which is position based).
>
>
> Q:\
> “Section 5.1 mentions that the particle density inputs are normalized for the pressure network. This is counterintuitive since density is an important feature for computing the pressure. Why is this necessary to stabilize the training?”
>
>
> A:\
> In practice, the particle density can range from 10-20 (varies depending on neighborhood radius and kernel function), this is quite large compared to other input like velocities. We scale them to be in the range of [0, 1] (using min-max scaler). Although their absolute magnitude is lost, their relative magnitude is still kept.
>
>
> Q:\
> “Will the code be released? If not it would be important to have a section in the appendix with all the important parameters for reproducing the networks in tabular form (layer sizes, normalizations, activations, etc.).”
>
>
> A:\
> We will make the code publicly available later on.

---

> > ### Comment · AnonReviewer4 · 2020-11-23
> > **More questions**
> >
> > Thank you for the answers. The updated version raises some more questions.
> >
> > - The canyon sequence shows the generalization well but why do the resting particles show some grid-like pattern?
> >
> > - Adding two more scenes for evaluation is good but this does not increase the test set a lot. As I understand the dam collapse and water fall scenes can be generated, so why not generate multiple instances for each scene type to reduce the variance?
> >
> > - The results for FGN have improved in Table 1. What changed?
> >
> > - I am not sure if I can follow this argumentation.
> >   > We have added inference time benchmark and model size comparison in Table 2. Small step size indeed decreases the difficulty for
> >   > inference on single frames, but we also observed that as more time steps are involved, accumulated position errors are higher for
> >   > end-to-end learning models (especially for CConv, which is position based).
> >
> >   In general smaller time steps lead to better accuracy but they also require more computation time. Therefore, it is of interest to find a timestep that is as large as possible that still satisfies the accuracy requirements. The timestep for FGN is very small. Can FGN work with larger timesteps as used in CConv? From the videos it looks like the CConv results are a lot worse compared to the original paper, therefore, it is not clear if the conclusions for end-to-end learning models hold in general.

---

> > > ### Author Response · Authors · 2020-11-23
> > > **Reply to the questions**
> > >
> > > Thank you for taking the time to read our updates on the paper.
> > >
> > >
> > > Q:\
> > > "The canyon sequence shows the generalization well but why do the resting particles show some grid-like pattern?"
> > >
> > > A:\
> > > The actual terrain of the Canyon model used here is quite complex (over 200k triangle meshes), and it contains small pits on the plain area which are not noticeable in the rendered video due to the resolution limitation. During the simulation, some particles will be stuck in these pits and thus exhibit a grid-like pattern.
> > >
> > >
> > > Q:\
> > > "Adding two more scenes for evaluation is good but this does not increase the test set a lot. As I understand the dam collapse and water fall scenes can be generated, so why not generate multiple instances for each scene type to reduce the variance?"
> > >
> > > A:\
> > > If the variance mentioned here is referring to the variance in Table 2 (single frame inference error on sub-networks), then we did generate multiple instances for each scene type to test the model. However, as we use very different viscosity parameter settings (0.1~0.0001), the variance of advection net is still high despite multiple instances are generated for each scene.
> > >
> > >
> > > Q:\
> > > "The results for FGN have improved in Table 1. What changed?"
> > >
> > > A:\
> > > Previously we have reported the performance of the model using one layer of aggregation in pressure net, we now updated it to the performance of using two layer of aggregation.
> > >
> > >
> > > Q:\
> > > "In general smaller time steps lead to better accuracy but they also require more computation time. Therefore, it is of interest to find a timestep that is as large as possible that still satisfies the accuracy requirements. The timestep for FGN is very small. Can FGN work with larger timesteps as used in CConv? From the videos it looks like the CConv results are a lot worse compared to the original paper, therefore, it is not clear if the conclusions for end-to-end learning models hold in general."
> > >
> > > A:\
> > > Unfortunately at this stage, FGN can not work with timestep as large as CConv (0.02s in their original paper). The main reason is, in FGN we adopt a force-based simulation scheme (advection based on gravity, viscosity, and then pressure projection), this scheme is very sensitive to the timestep (which is also the case for many force-based SPH methods)
> > >
> > > The worse performance of CConv in our test here is due to that in our training set we only include very simple geometries (simple cubic box, simple solid obstacles) and we did not sample different terrain in the training set (all boxes are smooth and flat). While in their original paper, they created multiple terrain geometries and included more fluid block shapes.\
> > > In the results of the original CConv paper,  there is still a noticeable difference between predictions from CConv and the ground truth sequence. We believe this is caused by the accumulation of error in every frame. Even though the single frame inference error can be small, these errors can still compound a large discrepancy of particle distribution in a long sequence.

---

### Official Review · AnonReviewer2 · 2020-10-28
**Very strong inductive bias for a very specific simulation type**

**Rating:** 4
**Confidence:** 4

**Review:**

The authors propose a learned model specialized on learning Lagrangian fluid dynamics for incompressible fluids. The model is a hybrid between a simulator with explicit advection, collision and pressure correction stages, and a learned model, trained by supervising each of those stages. The authors demonstrate improved stability/conservation of physical properties for a model, and some flexibility to the time-step being changed at test time.

The main general disadvantage I see is that the model seems oddly specific to the simulator used to generate the data, and specifically to incompressible fluids. The authors say “Although our model is entailed and customized for fluid simulation, it can be extended to simulation of other dynamics under Lagrangian framework as it takes universal feature“, but I am not sure how this work specifically supports extensions, since the main difference between the model and the baselines is precisely that while the baselines are general, this model specializes itself on one very specific type of simulation. Beyond this, there are some other important questions that I have listed below.

The advantages/claims are improved stability of divergence and density, and less learned parameters. The stability is not too surprising, since the model is built exactly to match the mechanisms that the simulator has to improve those, and in a sense it is using privileged information about the data generation. I would argue that it would have been easier to achieve the same in the baselines, just by adding additional loss components that explicitly minimize divergence and density error, using the same use of the privilege information. The smaller number of learned parameters is a nice result that potentially means the model can also run much faster than the baselines, so maybe this is the most important contribution: even assuming the model is meant to be specific for a domain, it can bring value by showing that it can be much faster than the baselines, and much faster than the ground truth simulator itself. Otherwise, I am not sure what the main selling point of a highly domain specific architecture is in this case.

All things considered, I am not sure the current state of the paper is sufficient to be accepted, both in terms of generality of the approach/main selling point, quality of the model description and comprehensiveness of the results. So my current rating is Reject (4) which I would be happy to immediately raise to Weak Reject (5) if the authors can confirm that they do not need access to intermediate simulator targets for training (Q1), and reconsider from there, given satisfactory explanations, additional evidence and discussion with other reviewers.

Main Comments/Questions:

1. The papers says:

> We train three networks, advection net, collision net and pressure net separately.

But also things like:

> the advection net predicts acceleration of particles (a^adv)
> The collision net takes … the relative position and velocity in intermediate state (let’s call this x*, v*)
> predicts correction to the velocity (Let’s call this \Delta v*, and the output v**)
> the updated intermediate position and velocity are take n as input by the pressure net (x*, v**0)
> predicts pressure (let’s call this p)

What are the inputs and targets of the advection net, the collision net and pressure net at train time, respectively? Does this means that apart from x^n and x^{n+1} your model needs access to all of the internal values of the simulator after each internal stage: (e.g., a^adv, x*, v*, v**, p), so you can supervise those when training each network separately. This would be a pretty strong requirement that the baselines do not require. How would you run this model on data obtained from a real experiment (assuming you had a way to do particle tracking, of course)? Or are in on the contrary the three models trained in order, always taking as inputs the outputs form the learned previous stage, and building the loss against the final position and velocity targets? The section around Equation (16) seems to imply that it is the former. If that is the case, improvements over baselines seem like a small gain considering the required access to all of that simulator-specific intermediate information.

2. In general the model description is quite hard to follow. For example, does the edge-focused and node-focused graph networks have a single step of message passing (later you talk about two layers of aggregation, but it is not clear how these two relate). Does the node focus graph network not embed the nodes features feature the aggregation? A lot of these decisions seem to deviate a bit from standard choices, and should probably be justified and explained more in detail.

How is \Nabla p (gradient of the pressure) calculated, do you actually take the gradients of the neural network with respect to its own input positions? Or do you use equation (19) to get a finite different estimate? Assuming it is the latter, is then the model not very sensitive to the connectivity radius of this network?

3. How is density error and velocity divergence defined? Specifically, in Figure 6, density deviation takes both positive and negative values, but in the table it takes only positive values.
Assuming this is calculated with equations (18) and (20), are these metrics not highly dependent on the neighborhood radius chosen? Is the neighborhood radius the same for the model aggregations than for the metrics? In that case, why is that?

Would it make sense to also report error as a function of the mse of the position somewhere, which has been the main metric in the baselines work?

4.
> we can observe oscillation on the free surface of fluids
> maintains a much more compact and smoother shape
> CConv fail to maintain smooth and compact fluid distributions
> GNS’ prediction is significantly slower than ground truth

It would also be very useful to be able to see some videos of the trajectories for the different models, to better understand the differences that the authors mention, rather than a single trajectory per dataset sampled at 5 points. Also, what does “slow” mean in this context?

5. In Fig 2(a) the first frame for GNS (t=100), seems very different from the rest, what is the reason for this? In similar DamCollapse domains in the GNS paper the accuracy seemed better somehow. Was the noise used to adjust the targets too, similar to the GNS model?

6. Any reason why the radius of connectivity is set differently for the advection/pressure net than for the collision net? Was the model sensitive to this?

7. Results section seems too short, only 16 lines worth of text, compared to two pages worth of tables and figures. I think more discussion is needed, and maybe more investigation of what specifically makes the model better. For example:

* Could the model be trained end to end, instead of training the three networks separately?

* The generalization section does not really elaborate enough about what the differences between training and generation datasets. The problem partly is that the dataset generation description itself is not very comprehensive, and seems hard to reproduce without access to the source code.

* The generalization to other time-steps should be studied in more detail, both how it was implemented, and what the results are for more both shorter and longer timesteps (rather than just the qualitative figure 3, with a single generalization time-step), and how specifically was the attempt to make it work for the baselines implemented.


Minor comments/typos (did not affect my decision):

“physic-informed” should be “physics-informed”?
“we aggregates” → (without s)
“are passed to processor” → “to the”
W in equation (13) is not defined in the text. Possibly add W next to: “using smooth
kernel as weight function”

The value for max density error, Dam collapse, ground truth, seems different in Table1 and Table2.

I would maybe put ground truth as the first row of the results tables for easier comparison to the first row, which has the strongest performance.

In Figure 6, could you make it so the color legend is consistent across plots, currently it is very confusing.

And in general the grammar needs a bit more work (especially in the 3.2 model section).

---

> ### Author Response · Authors · 2020-11-20
> **Responses to Reviewer1 (Part1)**
>
> We thank the reviewer for their time and detailed comments.
>
>
> Q:
> “The main general disadvantage I see is that the model seems oddly specific to the simulator used to generate the data, and specifically to incompressible fluids. ”
>
> A:
> The main reason we design the model structure specifically close to the ground truth simulator is based on the observation that pressure projection is a quite accurate and stable way to calculate pressure. Therefore the model is designed to mimic this scheme, and reduce the computational cost in the pressure projection step without losing much accuracy.\
> The whole model is customized for incompressible fluid simulation indeed, but we believe it can be extended to other particle system based on the following reasons:
> As we have shown in the revision version of the paper (in Table 2). Advection net and pressure net can act as solvers of two different dynamical systems respectively, even though they are using exactly the same network architecture. This indicates that the advection net is able to handle materials with different viscosity and body force. In addition, depending on the specific dynamics of the particle system, the pressure net can be removed or modified to predict other physical quantities.
>
>
> Q:
> “The smaller number of learned parameters is a nice result that potentially means the model can also run much faster than the baselines, so maybe this is the most important contribution: even assuming the model is meant to be specific for a domain, it can bring value by showing that it can be much faster than the baselines, and much faster than the ground truth simulator itself. ”
>
> A:
> We added an analysis of total trainable parameters and inference time benchmark in Table 3. It indicates that our model is much smaller than other baselines and significantly faster than both the ground truth solver and baselines. We also provide full detail of training set generation in the Appendix. (A.2. Dataset generation; Training strategy) It shows that we only use very simple training distribution (20 scenes, basic geometries).
>
>
> To Q1:\
> We do need to access the intermediate information during the simulation. \
> We have added a detailed illustration in the Appendix (A.2 Training Strategy) to show how we train three sub-networks separately (i.e. what data they used, how they are trained). \
> Inputs of advection net are x^n, v^n, nu (viscous parameter), and g. Target is a^adv.\
> Inputs of collision net are x^*, v^* (relative ones). Target is v^** - v^* (\Delta v).\
> Inputs of pressure net are v^**, rho (particle density). Target is p (pressure).\
> In general, this query of intermediate information does not cause much increase in the calculation time. We maintain two arrays (X and V) to track the positions and velocities of all particles. During the simulation, we query these arrays to calculate input for each network. a^adv and p are only used to update velocity, so we do not keep track of them during the whole process. Although this setting results in some loss of convenience, it enables the model to output more details of fluids (pressure distribution) and more importantly, significantly reduce the model size and training efforts needed (our training dataset only contains 20 trajectories).
>
>
> To Q2:\
> In the node-focused network, it has three steps of message passing in total. The first two steps use our version of aggregation (similar to graph SAGE) as the message passing function (which we defined as a single step of aggregation with two layers in the paper). After two steps, the node feature will be encoded as node embedding. The last step uses shared MLP as message passing function, the MLP takes node embedding on every single node as input. In preliminary tests, we found that adding a shared MLP in the last step of message passing brings an improvement in accuracy.\
> In the edge-focused network, it has two steps of message passing. The first step uses shared MLP as the message passing function, where the MLP is shared across every edge (which we defined as a single step of processing). In the last step, we use summation to aggregate all the edge embedding to their center nodes.\
>  \Nabla p (gradient of the pressure) is evaluated based on finite difference estimation. Indeed the model is not that sensitive to the radius of this network, it does not influence the single frame inference error very much, yet too small connectivity radius will result in penetration of particles in scenes that include complex geometries.

---

> > ### Author Response · Authors · 2020-11-20
> > **Responses to Reviewer1 (Part2)**
> >
> > To Q3:\
> > For each scene, we calculate the mean density of the fluid field in the initial frame (no update has been applied) as the constant density N0. In the following time steps, we calculate the mean density N_t. The error at time step t, is defined by: err_t = N_t - N0.
> > In the table, we report the absolute value of density error, but for the plot, we want it to reflect the full detail of error changing trend, so we keep the sign there.\
> > The velocity divergence is approximated by finite difference (equation (23) in the revised paper).\
> > It is true that the absolute magnitude of these metrics is highly dependent on neighborhood radius, however, their relative magnitude will not be affected by the chosen radius. Hence we can still use them to compare the performance of different models. We use a slightly smaller radius when evaluating metrics (2D).\
> > We added the MAE error for advection net and residual error (relative tolerance) for pressure net in the revised paper (Table 2).
> >
> >
> > To Q4:\
> > We have added a link for the video: https://sites.google.com/view/fluid-graph-network-video/home  \
> > In GNS’s result of the dam collapse simulation, the first collision between fluids and the right wall boundary happened about 50 frames later than the ground truth. Furthermore, in time step t=1000, for ground truth and FGN’s results, the fluids have already collided with the left wall boundary. However, in GNS’s result, the wave still has a distance to the left wall boundary.
> >
> >
> > To Q5:\
> > We do inject noise on both input and targets during the training of GNS model. This is very likely caused by GNS’s wrongly estimating viscosity. In our test case, the viscosity of fluids is not zero, this might affect the prediction of particles near the wall. As in GNS’s original paper, they also report failure cases of particles sticking to the wall. Moreover, our training set is smaller, which only contains 20 trajectories, which might also cause the model to give slightly worse performance. (GNS paper states they used 1000 trajectories for training)
> >
> >
> > To Q6:\
> > As collision net is used to predict collision and prevent particles overlapping on each other, we set the connectivity radius of it (0.9D) slightly smaller than particle size D. That means, only particles that are very close will be taken into consideration by the collision net.\
> > In general, for other networks, 10-20 neighbor particles is enough to generate realistic simulation. Increasing the radius will improve the accuracy a little bit (~2mm decrease in position loss when increasing radius from 3.0D to 4.0D), but will not affect the overall convergence.
> >
> > To Q7:\
> > We have now extended the Generalization part. We added simulation on two more complex scenes. We also added the position error trend figure under different time step sizes in the Appendix (Figure 10) and the position error trend of complex scenes (Figure 11).\
> > FGN,  GNS use the Euler integrator ( v^{n+1}=v^n + a^{n+1} dt, x^{n+1} = x^n + v^{n+1} dt) to integrate forward. To study the influence of different time step sizes, we change the dt in the above formula when testing them. As for CConv, we modify the output in the last layer to make it predict acceleration (similar to GNS) instead of position correction \Delta x, retrain the model, and then do the test.\
> > In addition, we have now provided full details of training dataset generation and training strategy. It would be much clearer to tell the difference between the test scene and the training scene.
> >
> >
> > Minor comments:
> >
> >
> > Q:\
> > “The value for max density error, Dam collapse, ground truth, seems different in Table1 and Table2.”\
> > A:\
> > Thanks for pointing out. In the previous version, Table 1 has some mistake values. We have corrected this in the revised version.
> >
> >
> > Q:\
> > “In Figure 6, could you make it so the color legend is consistent across plots, currently it is very confusing.”\
> > A:\
> > Thanks for the suggestion.
> > Legend has now been reformatted so that all plots hold a consistent color legend.

---

> > > ### Comment · AnonReviewer2 · 2020-11-20
> > > **Reviewer reply**
> > >
> > > Thank you for the comprehensive replies and updates to the paper. I really appreciate the updated notation, and additional model details.
> > >
> > > > The whole model is customized for incompressible fluid simulation indeed. We do need to access the intermediate information during the simulation.
> > >
> > > To me this really takes away from the interest of the paper. In its current form the model literally learns to distill the different internal components of MPS with very strong supervision on intermediate states, this is not mentioned either on the title or the abstract. I would be much more comfortable if the main claims of the paper were “Learning a fast approximation of MPS by distilling the three individual subcomponents into three different neural networks via supervised learning”, where the (much more limited) scope of the applicability of the paper is clearly stated.
> > >
> > > Similarly, I am not sure the comparisons to the baselines are fair. Not only neither of the two baselines (CConv or GNS) require use of any intermediate information, but also, the baselines do not assume access to velocity information, because in both cases v_t = x_t - x_tm1, (just changes in position). In this paper, v and x are not related that way, and v_t is truly part of the state. By not giving access the baselines to v_t, the baselines don’t have access to the full state unlike FGN, and if this is important, then the baselines would probably require to use separate position and velocity integrators, so the models have the ability to make independent predictions for x and v, like FGN can.
> > >
> > > A nice test of this, would be to force FGN to use v_t defined as x_t - x_tm1, e.g. in equation (14) use x^n instead of x*.
> > >
> > > > We added an analysis of total trainable parameters and inference time benchmark in Table 3.
> > >
> > > This has made me notice another apparent disadvantage of the model: by distilling the internal state of the solver, the model will never be able to run a larger time steps better than the ground truth simulator, right? In the paper you present generalization from a training time step of 0.002 to 0.004, however, this probably works only because the ground truth simulator would have also worked fine at 0.004, right?
> > > One of the advantages of the general learned simulator is that they usually can run at considerably larger timesteps than the ground truth simulator if you train them with larger timesteps. In this sense I am not sure the inference time compared to the baselines in Table 3 means much, since both the GNS and CConv baselines have a higher number of parameters and message passing steps to be more general, and be able to be trained with larger timesteps if necessary. Since FGN is very different in nature from the baselines, in order to compare time benchmarks, maybe it would be appropriate to use the most favorable timestep is for each model. (Of course this does not take away from the fact that it is faster than the ground truth simulator, which is indeed a valuable contribution).
> > >
> > > > we believe it can be extended to other particle systems based on the following reasons… even though they are using exactly the same network architecture...
> > >
> > > While the general idea of distilling subcomponents of a simulator is interesting, it is also not new (e.g. this is what most models are already doing when using an integrator in the loop, for example). So I am not sure how the specific contributions from this paper would be extended to other simulators beyond that general idea. I think the only technical contribution with respect to the neural networks architecture is maybe the proposed aggregator, but I do not think this is sufficient on its own to accept the paper in terms of generality. In that sense if the contribution is the aggregator, it would have been better to implement the aggregator in the context of the general-purpose baselines, and prove that it can improve the baselines in a more controlled setting.
> > >
> > > On the other hand, this would be a much more general model if the model trained it end to end, without access to the intermediate state, just imposing loss on the next step positions and velocities, and verify if this would still provide an advantage over baselines. Although even in that case, I would be concerned that maybe the inductive bias is too strong for most applications (e.g. only valid for incompressible fluids).
> > >
> > > > > Would it make sense to also report error as a function of the mse of the position somewhere, which has been the main metric in the baselines work?
> > > > We added the MAE error for advection net and residual error (relative tolerance) for pressure net in the revised paper (Table 2).
> > >
> > > Sorry, I meant that this would be an useful metric to compare to baselines too (e.g. Table 1), and to compare future models, in case you open source your datasets.

---

> > > > ### Comment · AnonReviewer2 · 2020-11-20
> > > > **Reviewer reply (2)**
> > > >
> > > > Some additional minor questions:
> > > >
> > > > Q1: what is the NNS time column, in Table 3?
> > > >
> > > >
> > > > Q2:
> > > > > We do inject noise on both input and targets during the training of GNS model.
> > > >
> > > > Could you specify how the noise added to the input position sequence relates to the noise injected to the targets (either with an example, or including the formula)?
> > > >
> > > > Q3: ”As collision net is used to predict collision and prevent particles overlapping on each other, we set the connectivity radius of it (0.9D) slightly smaller than particle size D”
> > > >
> > > > Am I understanding correctly that the collision net cannot compute messages unless the particles already have 10% overlap? Then how does the collision net solves complex collision situations that involve more than two particles. For example, assuming diameter D=1 picture particles A(x=0), B(x=1), C(x=1.5), D(x=2.5). With that connectivity, only B and C would be connected, but as soon as you solve the collision by moving them apart, then they will start colliding with A, and D, but because there aren’t connectivity edges, the model could not figure out that it also needs to move A and D in order to avoid collisions.

---

> > > > > ### Author Response · Authors · 2020-11-20
> > > > > **Author reply**
> > > > >
> > > > > Thank you for the helpful feedback.
> > > > >
> > > > >
> > > > > To the main concern:\
> > > > > In general, we design the model with these settings is not to replicate a specific solver (MPS here), but based on a very common scheme many force-based SPH methods adopted - advection and then pressure projection. Therefore we use subnetworks to learn and approximate these two processes to achieve better accuracy and improve calculation efficiency.\
> > > > > It is true that other general end-to-end learning models do not need to use internal states and thus are more convenient. Here we use internal information in order to approximate a higher-order integration, which noticeably improves accuracy without increasing model parameters.\
> > > > > Just like many other force-based SPH, the step size is very limited. As our model adopts a similar scheme (i.e. use forces to update fluids), unfortunately at this stage our model can only accept small step size, too.
> > > > >
> > > > >
> > > > > Q1:\
> > > > > It means the averaged Nearest Neighbor Searching time of every single frame.
> > > > >
> > > > >
> > > > > Q2:\
> > > > > Similar to the description in the Appendix of the original GNS paper. We draw noise from N(0, 0.0003). We add noise starting from the first input (v0, v1,v2, v3, v4, v5...).  Then at the second input, we add noise second times to some states (i.e. v1<-v1+e1+e2, e1 is added at the first time and e2 is added at the second time), and henceforth. Positions are adjusted so that positions are consistent with velocity.
> > > > >
> > > > >
> > > > > Q3:\
> > > > > It is true that this case will fail in our collision net setting. The main aim of adding the collision net into our model is not to accurately reproduce the collision system and repel all particles that are too close since pressure will keep most particles from getting too close. In practice, we found that about 100 particles in a 10k particle case will have a distance less than D (and usually for these close particles their distance will be even less than 0.75D), and the collision net is enough to drive them away.\
> > > > > In addition, we observe that when two particles come to a distance less than 1.0D, usually in the next pass through the pressure net, they will be driven away.

---

### Author Response · Authors · 2020-11-20
**Update in the Revision**

We thank all the reviewers for their time and insightful comments. Below is a summary of the update we made to the manuscript:


Main Update:

1. We uploaded videos of testing scenes, which can be found in https://sites.google.com/view/fluid-graph-network-video/home
2. We added a runtime analysis and model size comparison (Table 3).
3. We added two more tests on more complex geometries (Figure 5 and Figure 11 in the Appendix).
4. We extended the dataset generation detail and training strategy (i.e.how we get input and label for each sub-network) in the Appendix: A.2 Dataset generation and Strategy.
5. We added inference error analysis of a single network (advection net and pressure net) in Table 2.
6. We added the error trend figure of the model using different time step sizes in Figure 10 (in the Appendix).
7. We corrected some mistake data in Table 1.


Others:

* We modified the schematic in Figure 1 to make it more consistent with the model description.
* We changed notation for intermediate results in the Update Scheme part.
* We moved ablation analysis on different graph aggregators to Table 4.
* We changed the color legend in Figure 9 to make it easier to identify.
* We reorganized some paragraphs.

---

### Decision · Program_Chairs · 2021-01-07
**Final Decision**

**Decision:**

Reject

**Comment:**

The consensus recommendation is that the paper is not ready for publication at this time.